# ETS-MM: A Multi-Modal Social Bot Detection Model Based on Enhanced Textual Semantic Representation

Submission Id: 1033

## Abstract

Social bots are becoming increasingly common in social networks, and their activities affect the security and authenticity of social media platforms. Current state-of-the-art social bot detection methods leverage multimodal approaches that analyze various modalities, such as user metadata, text, and social network relationships. However, these methods may not always extract additional dimensions of semantic feature information that could offer a deeper understanding of users' social patterns. To address this issue, we propose ETS-MM, a multimodal detection framework designed to augment multidimensional information from text and extract the semantic feature representation of user text information. We first analyze the user's tweeting behavior based on topic preference and emotion tendency, integrating them into the textual data. Then, we try to extract enhanced semantic representations that reveal the latent relationship between tweeting behavior and tweet content while identifying potential contextual associations and emotional changes. Additionally, to capture the complex interaction between users, we integrate the user's multimodal information, including metadata, textual features, enhanced semantic features, and social network relationships to propagate and aggregate information across various modalities. Experimental results demonstrate that ETS-MM significantly outperforms existing methods across two widely used social bot detection benchmark datasets, validating its effectiveness and superiority.

## CCS Concepts

• **Computing methodologies** → **Natural language processing**; **Neural networks**; • **Information systems** → **Social networks**.

## Keywords

Social Bot Detection, Language Model, Large Language Model, Graph Neural Network

**ACM Reference Format:**

Anonymous Author(s). 2018. ETS-MM: A Multi-Modal Social Bot Detection Model Based on Enhanced Textual Semantic Representation. In *Proceedings of Make sure to enter the correct conference title from your rights confirmation emai (Conference acronym 'XX).* ACM, New York, NY, USA, 11 pages. https://doi.org/XXXXXXX.XXXXXXX

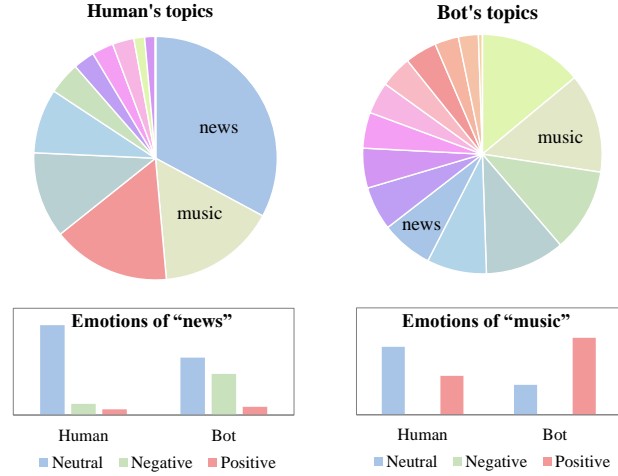

**Figure 1: Topic distribution of humans and bots in Cresci15 [9] and emotion distribution within the "news" and "music" topics.**

## 1 Introduction

Social bots often carry specific intentions, such as spreading disinformation [1, 10, 40, 44, 47], manipulating public opinion[7, 20, 21], promoting certain political or commercial agendas[3, 8, 11, 19, 38]. These bots pose a threat to the security of social media platforms. This threat affects users' trust and disrupts the overall information ecosystem[5, 35, 37]. Therefore, effectively detecting social bots has become a critical issue in social media. Researchers have shifted from single-modal to multimodal detection, integrating multiple data sources, such as user metadata, text, and social network relationships. By combining these diverse data sources, multimodal models can more comprehensively and accurately detect social bots[17]. Among these modalities, text, as the main information carrier, is not only the main form of interaction between users but also a key tool for them to influence their behavior and public opinion.

Users' text shows a huge difference between humans and bots regarding emotional expression and social interaction. In terms of social interactions, humans can understand and respond to the emotional states of others and establish deep social relationships. While bots lack an understanding of deep emotions and find it difficult to develop deep social relationships [4]. As shown in Figure 1, humans tend to have in-depth discussions around one or two topics, while bots cover a much wider range. Regarding emotional expression, bots can mimic human emotions in approximate situations. Still, humans have an advantage in the subtle differences in emotional

expression, while bots have difficulty replicating these subtle emotional differences[18]. This leads to differences in the distribution of emotions between bots and humans on specific topics. As shown in Figure 1, the emotion distributions for the "news" topic differ significantly between humans and bots; humans are more likely to express neutral or positive emotions, whereas bots tend to exhibit neutral or negative emotions.

Analyzing users' topics and emotions gives us a more comprehensive and accurate understanding of their behavioral patterns. However, current multimodal detection models tend to ignore the multidimensional semantic information in the text and the connection between this information when processing the text. Two main challenges are included: (1) Some works[6, 28] merely concatenate various users' text information into sequences without delving into the wealth of more dimension information contained within the tweets, such as topics and emotions. (2) Other works[14, 17, 32] directly use pre-trained LMs to obtain the encoding representations of tweets. This method lacks sophisticated mechanisms for extracting enriched semantic features, particularly those related to multidimensional information, which is crucial for distinguishing between bots and humans.

To address the above challenges, we propose a multimodal social bot detection framework incorporating topic preference and emotion tendency to enhance semantic feature representation. Specifically, we use a large language model (LLM) to identify user's most and least frequent topics and emotions to analyze the user's tweeting behavior, incorporating these insights into the tweet data. We then merge each user's description and tweet information (including tweets, topics, and emotions) into textual sequences. Subsequently, we combine each user's metadata with these textual sequences to create hybrid sequences. These hybrid sequences train a language model (LM), capturing enhanced semantic representations that reveal the relationships between topics and emotions within the textual data. Finally, we integrate the user's metadata, description, tweet features, enhanced semantic representation, and social network relationships using a graph neural network (GNN) to accurately determine whether a user is a bot or a human. This work makes the following contributions:

- We analyze the topic preference and emotion tendency in textual data to explore the differences between social bots' and humans' latent social behavior patterns.
- We capture enhanced semantic features that reveal relationships between tweeting behavior and content, identifying contextual associations and emotional changes.
- We integrate enhanced semantic features with multimodal features, enabling effective information propagation and aggregation across modalities, achieving SOTA in social bot detection.

## 2 Background

### 2.1 Multi-model Social bot Detection

Early methods for detecting social bots can be broadly categorized into feature-based, text-based, and graph-based. Feature-based approaches identify bots by extracting key attributes from user metadata and applying machine learning classifiers[27, 43, 45]. Text-based methods apply natural language processing techniques (NLP)

to analyze and encode users' textual content, primarily focusing on their tweets and descriptions, to identify social bots[15, 27, 42]. Graph-based methods detect bots by constructing social network relationships as graphs and applying GNNs for network analysis[2, 12, 26, 29, 34].

As social bots become increasingly sophisticated in their disguise capabilities, more researchers are employing multimodal approaches for social bot detection. Multimodal methods integrate multiple sources of information, such as user metadata, textual data, and social network information, for more comprehensive detection[24, 36, 46]. Feng et al.[17] enhanced the detection of bots with diverse camouflage behaviors by constructing a heterogeneous graph and exploiting multimodal user semantic and attribute information. Liu et al.[32] synthesized user representations from different prespectives by leveraging multimodal information (metadata, text, network structure). They introduced a modality-specific encoder and a community-aware expert hybrid layer to improve the accuracy and generalization of detection. Feng et al.[14] used a heterogeneous graph information network to learn node representations for graph-based detection of heterogeneity-aware bots, applying relational graph Transformers and semantic attention networks to account for user heterogeneity. Lei et al.[28] combined text and social network information, utilizing a text-graph interaction and semantic coherence approach to assess and counteract bots' evolving behaviors comprehensively. Cai et al.[6] represented each user as a text sequence and performed domain adaptation through an LM, using the model's output as the input features for the GNN. They distilled knowledge from the GNN back into the LM, enhancing the robustness of social bot detection.

Compared to methods that rely on a single source of information, such as user metadata, text, or social network relations, multimodal information provides a more comprehensive analysis.

### 2.2 Text Processing in Social Bot Detection

The processing of textual modalities is a key aspect of social bot detection. Early methods typically use NLP techniques to extract features from text to identify bots. Kudugunta et al.[27] proposed a long short-term memory(LSTM)-based deep network model to process the tweet and account levels. Wei et al.[42] used a BiLSTM modal to detect Twitter bots for featureless engineering.

Current research has concatenated user data into text sequences and used LMs to obtain the encoded representations[6, 28]. However, these methods fail to extract more dimension information within the tweets. Other approaches to text processing in multimodal methods encoded users' textual information using pretrained LM. Feng et al.[17] obtained the user descriptions and embedded representation of tweets through an LM, averaging the encoded representation of each tweet to get an overall representation of the user's tweets. However, this approach widely used in social bot detection[14, 17, 32] does not include advanced techniques for extracting semantic features, especially those involving multidimensional information.

Although multimodal social bot detection models effectively recognize bots, they often fail to capture textual information. Therefore, we propose extracting the topics and emotions from user tweets through LLM. By leveraging the knowledge of LLM, we can obtain

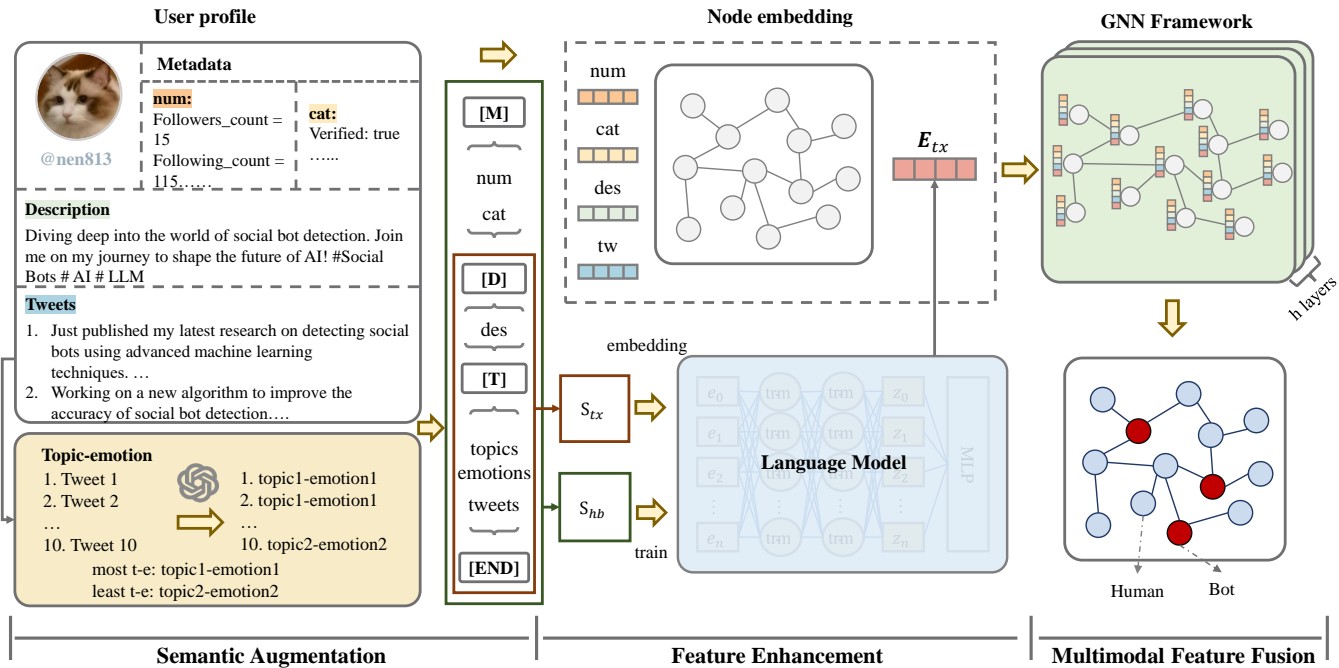

**Figure 2: The framework of ETS-MM.**

a more comprehensive representation of the tweets, capturing both expressive and contextual features. These topics and emotions are integrated into the user's information, which is then used to train the LM. This process represents the user's textual information (including descriptions, topics, emotions, and tweets) more holistically, ensuring a deeper and more comprehensive understanding of the user's textual behavior.

## 3 Methodology

We present the ETS-MM framework that consists of three modules: the semantic augmentation module, the feature enhancement module, and the multimodal feature fusion module, as shown in Figure 2. The semantic augmentation module uses an LLM to extract topics and emotions to augment the text data. The feature enhancement module extracts the metadata, textual, and enhanced semantic features. The multimodal feature fusion module employs a GNN to integrate multimodal features, including user metadata, textual information, enhanced semantic features, and social network relationships.

### 3.1 Semantic Augmentation

To extract more dimensional semantic information, we construct two types of sequences: textual sequences and hybrid sequences.

**Textual sequences**. The text sequence $S_{tx}$ includes the user's description, tweets:

$$S_{tx} = [Description]\{description\}[Tweet]\{tweet\}[END] \quad (1)$$

where tweets include topics and emotions extracted from LLM, following Chang et al. [8]. We extract their ten longest tweets by

GPT-3.5 [30] to identify each tweet's corresponding topic preference and emotion tendency. Appendix A.1 shows the prompts for extracting topics and emotions from GPT-3.5.

- **Topic preference**. We identify the subjects that the user frequently engages within their tweets. This involves categorizing tweets based on content and determining the most and least frequent topics $topic_{most/least}$. The topics are drawn from a list of 16 Twitter categories[1].
- **Emotion tendency**. We further analyze the emotion tendency corresponding to each tweet topic by identifying the most and least frequent emotions $emotion_{most/least}$. The emotions are classified based on Plutchik's three sentiment categories[19].

To further enhance the tweets representation, we incorporate a sequence of topics and emotions into the tweet sequence representation: $\{topic - emotion\}$ = *Most of these tweets are about $topic_{most}$ and $emotion_{most}$, a few of these are about $topic_{least}$ and $emotion_{least}$.* We add this sequence into the sequence of tweets:

$$\{tweet\} = [Tweet]\{topic - emotion\}[SEP]\{tweet_1\}[SEP]$$
$$\{tweet_2\}[SEP]...[SEP]\{tweet_n\} \quad (2)$$

where $tweet_i$ represents the $i$-th tweet, and $n$ represents the overall count of user tweets.

**Hybrid sequences**. The hybrid sequence $S_{hb}$ includes user metadata and text sequence $S_{tx}$:

$$S_{hb} = [Metadata]\{metadata\}[Description]\{description\}$$
$$[Tweet]\{tweet\}[END] \quad (3)$$

---

[1]https://inboundfound.com/twitter-topics-list/

Numerical feature          Categorical feature

**[Metadata]**15**[SEP]**115**[SEP]**256**[SEP]**61**[SEP]**8**[SEP]**1**[SEP]**
**[Description]**Diving deep into the world of social bot detection. Join me on my journey to shape the future of AI! #Social Bots # AI # LLM
**[Tweet]**Most of these tweets are about news and positive, a few of these are about food and neutral.**[SEP]**Just published my latest research on detecting social bots using advanced machine learning techniques. **[SEP]**Working on a new algorithm to improve the accuracy of social bot detection**[SEP]**…**[END]**

$S_{hb}$

**[Description]**Diving deep into the world of social bot detection. Join me on my journey to shape the future of AI! #Social Bots # AI # LLM
**[Tweet]**Most of these tweets are about news and positive, a few of these are about food and neutral.**[SEP]**Just published my latest research on detecting social bots using advanced machine learning techniques. **[SEP]**Working on a new algorithm to improve the accuracy of social bot detection**[SEP]**…**[END]**

$S_{tx}$

**Figure 3: The example of a hybrid sequence and a textual sequence.**

where $\{metadata\}$ is the sequence of user metadata:

$$\{metadata\} = [Metadata]\{metadata_1\}[SEP]\{metadata_2\} \\ [SEP]...[SEP]\{metadata_m\} \quad (4)$$

where $metadata_i$ represents numerical and categorical features of users, and $m$ represents the overall count of metadata features. Figure 3 shows the example of a hybrid and textual sequence.

### 3.2 Feature Enhancement

We extract metadata, textual, and enhanced semantic features for node embedding.

**Metadata Feature**. Following the feature extraction approach by Feng et al.[17], we process metadata, description, and tweet features. Metadata refers to the user's numerical and categorical features, $num^{(0)}$ and $cat^{(0)}$.

**Textual Feature**. The description representation $des^{(0)}$ is obtained by a pre-trained LM model. For the tweet representation $twe^{(0)}$, we first use the LM model to obtain the embedding for each tweet and then compute the average embedding of all tweets.

**Enhanced Semantic Feature**. We train LM with hybrid sequences $S_{hb}$ to capture the enhanced semantic features from the user's augmented semantic data. We design our training framework from the LMs proposed by Cai et al.[6]. Firstly, we use a pre-trained LM model to obtain the user representation $z_i$ and apply mean pooling to the LM output:

$$z_i = \frac{1}{M_i} \sum_{j=1}^{M_i} LM(S_{hbi})_j \quad (5)$$

where $S_{hbi}$ is $i$ - th user's hybrid sequence. $LM(\cdot)$ is a text encoder to obtain the representation of $S_{hb_i}$, and $M_i$ is the number of tokens in $S_{hb_i}$. We obtain a 768-dimensional embedding for the user representation.

We utilize an $L$ layer MLP to perform feature dimensionality reduction and project it into a binary classification space to determine whether the user is human or bot. The final predictions are computed using the $Softmax$ function:

$$\hat{y}_i = Softmax(LeakyReLU(W_{(l)} \cdot z_i^{(l-1)} + b_{(l)})) \quad (6)$$

where $W_{(l)}$ and $b_{(l)}$ are learnable parameters. The model is then optimized using the cross-entropy loss function.

Finally, the enhanced semantic representation $E_{tx}$ within the relationships between topics and emotions is derived by embedding the textual sequences $S_{tx}$ using the trained LM:

$$E_{ts} = LM(X; \theta^*), X \in S_{tx} \quad (7)$$

where $\theta^*$ denotes the trained LM parameters.

### 3.3 Multimodal Feature Fusion

We integrate the user's metadata feature, textual feature, social relationships, and enhanced semantic features using a GNN to classify the user. We construct the social relationships between users as a graph and apply the GNN to learn user representations. The structure of the GNN framework is illustrated in Figure 4. Specifically, we first transform the user's features into node embeddings by passing them through a linear layer followed by ReLU activation (linear_relu layer):

$$num_i^{(h)\prime} = ReLu(Linear(W_{(h)}^0 \cdot num_i^{(h)} + b_{(h)}^0)) \\ cat_i^{(h)\prime} = ReLu(Linear(W_{(h)}^1 \cdot cat_i^{(h)} + b_{(h)}^1)) \\ des_i^{(h)\prime} = ReLu(Linear(W_{(h)}^2 \cdot des_i^{(h)} + b_{(h)}^2)) \quad (8) \\ twe_i^{(h)\prime} = ReLu(Linear(W_{(h)}^3 \cdot twe_i^{(h)} + b_{(h)}^3)) \\ E_{tx_i}^{(h)\prime} = ReLu(Linear(W_{(h)}^4 \cdot E_{tx_i}^{(h)} + b_{(h)}^4))$$

where $W_{(h)}^j$ and $b_{(h)}^j$ are the learnable parameters of the $h$-layer of the GNN convolution (Gconv), where $j$ is an integer between 0 and 4. $i$ represents the linear layer that processes different user features. Afterward, we concatenate all the five user features to form the final representation $x_i^{(h)}$:

$$x_i^{(h)} = num_i^{(h)\prime} \oplus cat_i^{(h)\prime} \oplus des_i^{(h)\prime} \oplus twe_i^{(h)\prime} \oplus E_{tx_i}^{(h)\prime} \quad (9)$$

The fused features are passed through a linear_relu layer, then combined with the graph-based user relationship features to create

**Figure 4: GNN architecture for multimodal Feature Fusion.**

a unified representation:

$$a_i^{(h+1)} = \underset{j \in \mathcal{N}(i)}{AGGER}(x_i^h, x_j^h, e_{ij}) \qquad (10)$$

$$x_i^{(h+1)} = ReLu(Linear(UPDATE(x_i^h, a_i^h))) \qquad (11)$$

where $AGGER(\cdot)$ represents the information aggregated from the neighboring users of user $i$. $\mathcal{N}(i)$ indicates the set of neighbors of user $i$. $e_{ij}$ denotes the edge between user $i$ and user $j$. $UPDATE(\cdot)$ denotes updating the user's representation based on the aggregated information. This updated information is passed through a linear_relu layer to obtain the user representation in layer $h + 1$. Finally, the GNN is optimized using a cross-entropy loss function.

## 4 Experiment

In this section, we aim to answer the following question in this experiment:

- **Q1**: How does the ETS-MM method perform compared to other multimodal-based social bot detection methods?
- **Q2**: Which combinations of LMs and GNNs can significantly enhance the accuracy of social bot detection?
- **Q3**: Can topics and emotions extracted from LLMs improve the effectiveness of social bot detection frameworks?
- **Q4**: Can enhanced semantic features improve the performance of social bot detection tasks?
- **Q5**: How do different modalities influence the model's performance?

### 4.1 Experiment settings

*4.1.1 Datasets.* In our experiments, we use two datasets: Cresci15[9] and Twibot20[16]. Cresci15 includes metadatas, descriptions, tweets, and social network information of humans and bots, making it a classic dataset in social bot detection. Twibot20 is larger in scale, providing a broader range of scenarios and diverse user samples. More dataset statistics are outlined in Appendix A.2.

*4.1.2 Baselines.* Since we utilizes data from various sources, the focus is on comparing it with multimodal-based social bot detection methods. HGT[25] and RGT[14] concentrate on processing heterogeneous graph. SimpleHGN[33] demonstrates that simple isomorphic graph GNNs can achieve high effectiveness when configured appropriately. BotRGCN[17] integrates relational graph convolutional networks with multimodal information to improve detection performance. BIC[28] highlights the deep interaction between text and graphs, along with semantic consistency modeling. BotMoE[32] focuses on multimodal information fusion and incorporates a community-aware mixture of experts. Lastly, LMBot[6] realizes graph-independent bot detection through knowledge distillation of LMs.

*4.1.3 Model Parameters.* In our model, we use several parameters to achieve optimal performance, as shown in Table 1.

**Table 1: Parameter settings of ETS-MM.**

| Parameter | Value |
|---|---|
| Batch size in training LM task | 32 |
| Epochs of training LM | 3 |
| Max length of LM | 512 |
| Hidden dimensions of LM | 128 |
| Dropout of LM | 0.1 |
| Warmup of LM | 0.6 |
| Learning rate of LM | 1e-5 |
| Weight decay of LM | 0.01 |
| Epochs of training GNN | 300 |
| Number of GNN convolution layers | 2 |
| Dropout of GNN | 0.5 |
| Learning rate of GNN | 1e-3 |
| Weight decay of GNN | 1e-4 |
| Activation of GNN | Leakyrelu |
| Optimizer of LM and GNN | Adamw |

### 4.2 Performance of ETS-MM(Q1)

Our experiments demonstrate that the ETS-MM model consistently outperforms other state-of-the-art social bot detection methods on Cresci15 and Twibot20, as detailed in Table 2. Specifically, on the Cresci15 dataset, ETS-MM achieves an accuracy of 99.10%, exceeding the second-ranked LMBot. Similarly, on Twibot20, ETS-MM has an accuracy of 90.05%, ahead of BotMoE's. Notably, while LMBot performs well on the smaller Cresci15 dataset, its accuracy drops significantly on the larger Twibot20 dataset, likely due to incomplete semantic feature extraction on the larger dataset. ETS-MM demonstrates consistent and stable performance across multiple

**Table 2: Average accuracy and F1-socre of five runs on Cresci15 and Tweibot 20 datasets, with standard deviations in parentheses, best performance bold and second-best underlined.**

| Method | pre-train LM | train LM | train GNN | Cresci15 | | Twibot20 | |
|---|---|---|---|---|---|---|---|
| | | | | Accuracy | F1-score | Accuracy | F1-score |
| HGT[25] | | | ✓ | 97.45(±0.23) | 96.87(±0.43) | 86.56(±0.43) | 88.75(±0.67) |
| SimpleHGN[33] | | | ✓ | 96.84(±0.13) | 97.44(±0.74) | 87.89(±0.23) | 89.56(±0.31) |
| BotRGCN[17] | | | ✓ | 96.52(±0.71) | 97.30(±0.53) | 83.27(±0.57) | 85.26(±0.38) |
| RGT[14] | | | ✓ | 97.15(±0.32) | 97.78(±0.24) | 86.57(±0.41) | 88.01(±0.41) |
| BIC[28] | | ✓ | ✓ | 98.35(±0.24) | 98.71(±0.18) | 87.61(±0.21) | 89.13(±0.15) |
| BotMoE[32] | | | ✓ | 98.50(±0.00) | 98.82(±0.00) | 87.76(±0.2) | 89.22(±0.3) |
| LMBot[6] | ✓ | ✓ | ✓ | 99.06(±0.30) | 99.26(±0.20) | 85.25(±0.20) | 87.38(±0.20) |
| ETS-MM | | ✓ | ✓ | **99.10(±0.08)** | **99.29(±0.06)** | **90.05(±0.19)** | **90.82(±0.18)** |

runs, with a standard deviation of only ±0.19% and ±0.18% on Twibot20, highlighting the robustness of the model's improvements. Interestingly, training GNNs using only extracted user metadata and textual features, without additional LM training, led to inferior results. This indicates that LM training effectively captures crucial semantic features in textual information. Enhancing semantic features proves crucial. Almost all models trained with LMs and enriched semantic features perform better, particularly ETS-MM, which reached SOTA on both datasets after semantic enhancement. This confirms the importance of extracting topic preference and emotion tendency in social bot detection.

### 4.3 LM and GNN Impact on Detection(Q2)

This experiment examines the effect of different combinations of LMs and GNNs on social bot detection performance. We select four LMs–BERT[13], DeBERTa[23], RoBERTa-f[2] and RoBERTa[31]–and four GNNs–GAT[41], HGT[25], SAGE[22], and RGCN[39]. The results indicate that various combinations of LMs and GNNs significantly influence the model's performance, as shown in Table 3.

The Cresci15 dataset shows the most consistent BERT performance and the best RoBERTa results. Since Cresci15 is simpler compared to Twibot20, it may benefit more from the lighter model combinations. In the Twibot20 dataset, RoBERTa and RoBERTa-f outperform BERT and DeBERTa, possibly because RoBERTa generated the original node embeddings, allowing it to analyze complex data structures better. Specifically, in the Cresci15 dataset, the combination of RoBERTa and GAT achieves the best performance, followed by RoBERTa combined with RGCN. This suggests that GAT, with its simplified graph structure aggregation, is more efficient at capturing key features. In contrast, overly complex GNN structures like HGT and RGCN may introduce noise in this scenario. For the Twibot20 dataset, the combination of RoBERTa-f and SAGE performs the best, significantly surpassing other combinations. This implies that SAGE is particularly effective at leveraging the enhanced semantic features extracted by RoBERTa-f to capture the complex relational networks of social bots. Similarly, the combination of RoBERTa and GNN is the most stable, with the model successfully utilizing relational features between users.

[2]https://huggingface.co/yzxjb/roberta-finetuned-20

### 4.4 Effectiveness of Topics and Emotions(Q3)

To assess the impact of topics and emotions on social bot detection, we design four sets of experiments: without topics and emotions(*no_te*), with only topics(*t*), with only emotions(*e*), and with both topics and emotions(*te*). We analyze the accuracy and F1-score of the four sets.

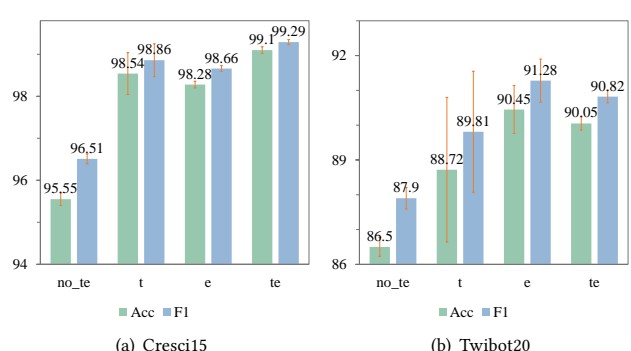

(a) Cresci15          (b) Twibot20

**Figure 5: Accuracy, F1-score and standard deviation of ETS-MM across four different scenarios: without topics and emotions(*no_te*), with only topics(*t*), with only emotions(*e*), and with both topics and emotions(*te*).**

The results of the experiments are shown in Figure 5. In both datasets, adding only topics and emotions improves detection performance. This is because topic information helps capture the potential behavioral patterns of humans and bots, and emotion information allows for the analysis of users' emotional changes and diversity, thus improving the model's performance. By examining the topic distribution in the Cresci15 dataset, as shown in Figure 1, we can clearly distinguish between humans and bots: humans prefer to focus on one or two topics. In contrast, bots have a much broader topic distribution. In Cresci15, the distribution of emotions within topics also varies. For example, in the "news" and "music" topics, there is a significant difference in the emotional distribution between humans and bots. By combining the topics and emotions, richer semantic features are extracted. However, in Twibot20, the model's performance slightly declined. This may be because not all topics and emotions can provide sufficient information for the

**Table 3: Accuracy, F1-score and the standard deviations for combinations of four LMs(BERT, DeBERTa, RoBERTa-f and RoBERTa) and four GNNs(GAT, HGT, SAGE and RGCN) on Cresci15 and Twibot20.**

| Dataset | Cresci15 | | | | | | | |
|---|---|---|---|---|---|---|---|---|
| Model | BERT | | DeBERTa | | RoBERTa-f | | RoBERTa | |
| | Accuracy | F1-score | Accuracy | F1-score | Accuracy | F1-score | Accuracy | F1-score |
| GAT | 98.47(±0.08) | 98.80(±0.06) | 97.72(±0.16) | 98.22(±0.11) | 98.06(±0.36) | 98.48(±0.28) | **99.14(±0.36)** | **99.32(±0.28)** |
| HGT | 98.47(±0.08) | 98.80(±0.06) | 97.68(±0.17) | 98.19(±0.13) | 97.42(±0.16) | 98.00(±0.11) | 98.5(±0.00) | 98.83(±0.00) |
| SAGE | 98.50(±0.00) | 98.83(±0.00) | 97.57(±0.00) | 98.11(±0.00) | 97.91(±0.24) | 98.37(±0.18) | 98.95(±0.31) | 99.18(±0.24) |
| RGCN | 98.50(±0.00) | 98.83(±0.00) | 97.91(±0.08) | 98.35(±0.07) | 97.50(±0.28) | 98.05(±0.21) | 99.10(±0.08) | 99.29(±0.07) |
| Dataset | Twibot20 | | | | | | | |
| Model | BERT | | DeBERTa | | RoBERTa-f | | RoBERTa | |
| | Accuracy | F1-score | Accuracy | F1-score | Accuracy | F1-score | Accuracy | F1-score |
| GAT | 83.99(±1.81) | 85.71(±1.99) | 85.21(±0.49) | 86.79(±0.43) | 87.86(±2.42) | 88.97(±2.26) | 88.76(±0.34) | 89.77(±0.29) |
| HGT | 86.64(±0.34) | 87.94(±0.34) | 86.97(±0.20) | 88.29(±0.11) | 90.96(±0.20) | 91.63(±0.19) | 90.08(±0.19) | 90.85(±0.18) |
| SAGE | 87.17(±0.33) | 88.58(±0.24) | 87.12(±0.30) | 88.40(±0.23) | **91.21(±0.28)** | **91.93(±0.27)** | 90.16(±0.18) | 91.00(±0.13) |
| RGCN | 86.48(±0.32) | 87.91(±0.32) | 87.13(±0.22) | 88.31(±0.23) | 91.17(±0.22) | 91.85(±0.21) | 90.05(±0.19) | 90.82(±0.19) |

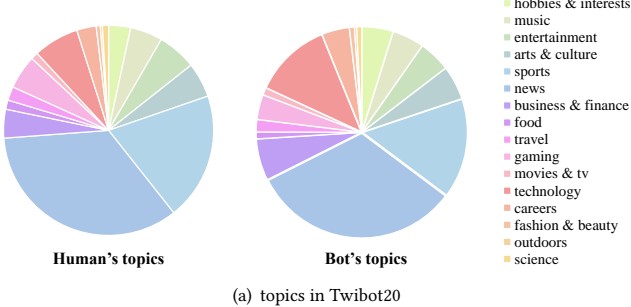

(a) topics in Twibot20

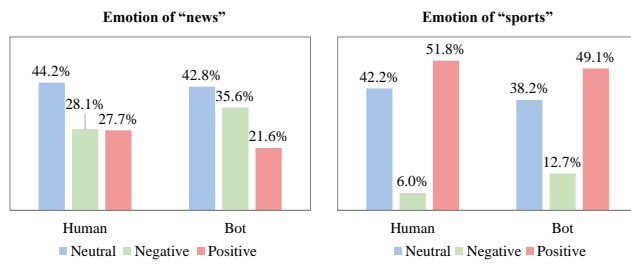

(b) emotions in Twibot20

**Figure 6: Topic distribution of humans and bots in Twibot20 and emotion within the "news" and "sports" topics.**

detection model. For example, as shown in Figure 6, some emotions (such as "neutral") or highly general topics (such as "news") may not be distinctive enough, thus contributing finitely to the model's detection capability. Twibot20's topic distribution difference between humans and bots is smaller than Cresci15. This phenomenon further demonstrates that bots' ability to disguise themselves continuously improves. Appendix A.3 further enumerates the confusion matrices

of the model for four cases: without topics and emotions, only with topics, only with emotions, and with topics and emotions.

## 4.5 Contribution of Enhanced Semantic Features(Q4)

This subsection investigates the contribution of enhanced semantic features to the social bot detection framework. We design two different scenarios: without enhanced semantic features ($E_{tx}$) and with enhanced semantic features ($E_{tx}$). In our experiments, we compared the feature distributions of the final layer of the ETS-MM model under two conditions across both datasets using t-SNE visualization, as shown in Figure 7.

The model displays a much clearer clustering structure when enhanced semantic features were included. This effect is particularly evident in the Cresci15 dataset, where the separation between bots (pink dots) and humans (blue stars) becomes more distinct. Several factors contribute to this improvement: First, enhanced semantic features, such as topics and emotions, allow for a more comprehensive capture of contextual associations and subtle emotional changes within the text. This gives the model a richer semantic background, enabling it to detect behavioral patterns and characteristics specific to social bots, thereby improving detection accuracy. In contrast, without the enhanced feature, the model may rely solely on basic text representations like word vectors or simple sentence embeddings, which are often insufficient for capturing the deeper patterns of user behavior. The model can combine these multidimensional elements by integrating topics and emotions, creating a more comprehensive representation of each user. Appendix A.4 shows the model's performance in two cases: without enhanced semantic features and with enhanced semantic features.

## 4.6 The Influence of Different Modalities(Q5)

To evaluate the impact of different modalities on the performance of our model, we design four experimental settings: (1) using metadata

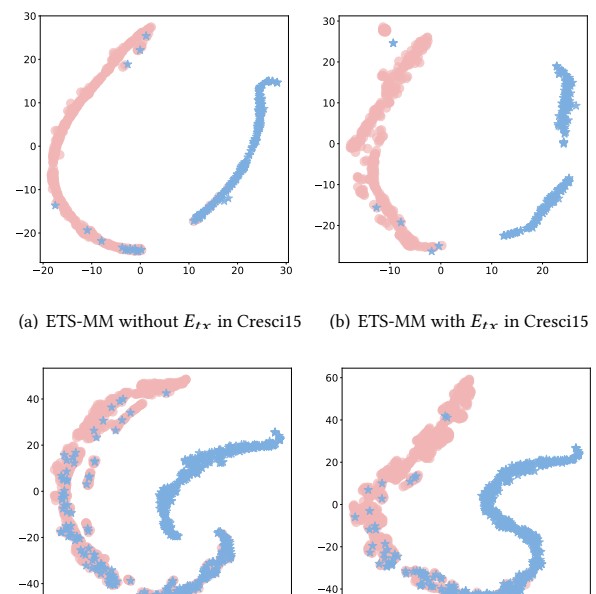

(a) ETS-MM without $E_{tx}$ in Cresci15    (b) ETS-MM with $E_{tx}$ in Cresci15

(c) ETS-MM without $E_{tx}$ in Twibot20    (d) ETS-MM with $E_{tx}$ in Twibot20

**Figure 7: The feature distributions were obtained by applying t-SNE to reduce the dimensionality of output features of the model's final layer across two different scenarios. The first two figures show the results from the Cresci15 dataset, while the last two represent the Twibot20 dataset. Pink dots denote bots, and blue stars indicate humans.**

and textual features, (2) using metadata and social network features, (3) using textual and social network features, and (4) using all features (metadata, text, social network), where metadata features include numerical and categorical features, textual features include description representation, tweet representation, and enhanced semantic features.

As shown in Table 4, removing text features led to the most significant drop in performance, indicating the critical role of textual information in detecting social bots. Removing metadata also resulted in a performance decline, but the impact was less severe. Interestingly, removing metadata features had a smaller impact on the Cresci15 dataset, likely due to its simpler data structure. Removing each modality in Twibot20 has a significant effect on the results. In addition, we extracted five types of features for generating user embeddings across two GNN convolution layers and visualized them using t-SNE, as shown in Figure 8. The t-SNE visualization clearly shows that different features contribute to distinct clustering patterns. In both datasets, these modalities integrate with one another. Particularly in the Cresci15 dataset, feature points from different modalities gradually merge into the same category cluster, further validating the effectiveness of our proposed method.

**Table 4: Impact of different modalities (metadata, text, and social network) on model performance across Cresci15 and Twibot20 datasets.**

| Meta | Text | Network | Cresci15 | |
|------|------|---------|------|------|
| | | | Acc | F1 |
| ✓ | ✓ | | 95.96(±0.49) | 96.86(±0.36) |
| ✓ | | ✓ | 92.15(±0.78) | 93.99(±0.58) |
| | ✓ | ✓ | 98.28(±0.24) | 98.65(±0.19) |
| ✓ | ✓ | ✓ | **99.10(±0.08)** | **99.29(±0.06)** |

| Meta | Text | Network | Twibot20 | |
|------|------|---------|------|------|
| | | | Acc | F1 |
| ✓ | ✓ | | 83.63(±0.39) | 85.34(±0.47) |
| ✓ | | ✓ | 81.52(±0.22) | 85.39(±0.16) |
| | ✓ | ✓ | 82.38(±0.56) | 83.75(±0.63) |
| ✓ | ✓ | ✓ | **90.05(±0.19)** | **90.82(±0.18)** |

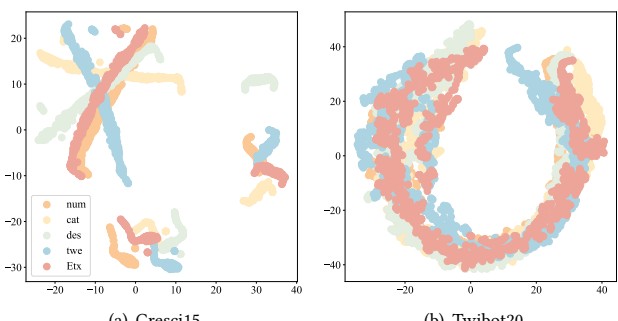

(a) Cresci15      (b) Twibot20

**Figure 8: The feature distributions of five features($num$, $cat$, $des$, $tw$ and $E_{tx}$) across two GNN convolutional layers were visualized using t-SNE.**

## 5 Conclusion

In this study, we propose ETS-MM, a multimodal bot detection model that enhances the semantic representation of user text and addresses the issue of textual modality lacking more dimensional information. Using LLM, we extract topic preference and emotion tendency from tweets, integrating these insights into two custom sequences: textual sequences and hybrid sequences, which combine user metadata and augmented text. These sequences help train the LM and encode enhanced semantic representations. Additionally, GNN is employed to integrate various features, ultimately identifying bots. Our experiments show that the ETS-MM model outperforms existing methods in social bot detection. In the future, we will explore a closer integration of large language models for social bot detection and extract more effective information from user text.

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

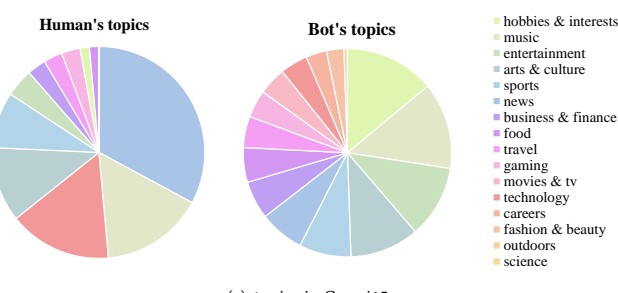

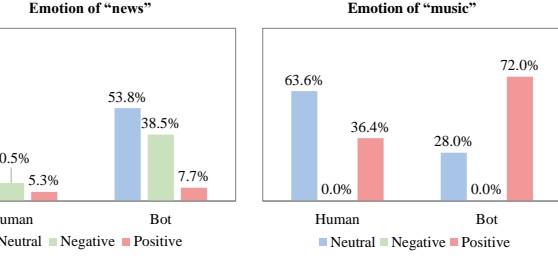

(a) topics in Cresci15

Emotion of "news"

84.2%
10.5%  5.3%
53.8%  38.5%  7.7%
Human    Bot
Neutral  Negative  Positive

Emotion of "music"

63.6%  36.4%
0.0%
72.0%
28.0%  0.0%
Human    Bot
Neutral  Negative  Positive

(b) emotions in Cresci15

**Figure 9: Topic distribution of humans and bots in Cresci15 and emotion distribution within "news" and "music" topics.**

## A APPENDIX

### A.1 Prompts for Extracting Topics and Emotions from ChatGPT

We reference [8] to design the instruction prompts we extract for topics and emotions for the construction of the textual dataset in Section 3.1, as shown in Table 5.

**Table 5: The instruction prompt and an example of the output of GPT-3.5-Turbo**

| Instruction prompt | Example output |
|---|---|
| Please classify each tweet into the topics and corresponding emotions. The available topics are arts & culture, business & finance, careers, entertainment, fashion & beauty, food, gaming, hobbies & interests, movies & TV, music, news, outdoors, science, sports, technology, and travel. The emotions to consider are positive, negative, and neutral. Please provide the classification for each post in the format 'topic - emotion'. If you are not sure about the 'topic' corresponding to this tweet, classify the 'topic' as none. Limit the response to less than 100 words and use lowercase. | 1: news - positive
2: news - negative
3: news - positive
4: news - neutral
5: news - positive
6: news - positive
7: news - positive
8: news - negative
9: news - neutral
10: news - neutral |

### A.2 Statistics of Datasets

Table 6 shows the number of training sets, validation sets, test sets, and overall in the Cresci15 and Twibot20 datasets.

**Table 6: Statistics of Datasets.**

| Datasets | train | dev | test | total |
|---|---|---|---|---|
| Cresci15 | 3708 | 1058 | 535 | 5301 |
| Twibot20 | 8278 | 2365 | 1183 | 11826 |

### A.3 Confusion Matrices of ETS-MM with Topics and Emotions

Figure 10 shows the confusion matrices of the model for the four cases: without topics and emotions, with only topics, with only emotions, and with both topics and emotions in Section 4.4. On the Cresci15 dataset, adding topics and emotions makes the model able to distinguish humans more clearly, especially topics. This also shows that topics are important in distinguishing bots from humans. On the Twibot20 dataset, the model's accuracy for predicting humans and bots increases. This demonstrates that emotions are also beneficial for improving social bot detection.

### A.4 Results of Enhanced Semantic Features

Table 7 illustrates the performance changes in two different scenarios: without enhanced semantic features $E_{tx}$ and with enhanced semantic features $E_{tx}$, in Section 4.5. The performance changes are illustrated in Table 7. In the Cresci15 and Twibot20 datasets, after removing enhanced semantic features, the model's accuracy decreased by 3.55% and 2.78%, while the F1-score dropped by 3.55% and 2.92%.

**Table 7: Accuracy, F1-score, and standard deviation of ETS-MM across two different scenarios: without and with enhanced semantic features.**

| Methods | Cresci15 | |
|---|---|---|
|  | Accuracy | F1-score |
| without $E_{tx}$ | 95.55(±0.15) | 96.51(±0.12) |
| with $E_{tx}$ | **99.10(±0.08)** | **99.29(±0.06)** |

| Methods | Twibot20 | |
|---|---|---|
|  | Accuracy | F1-score |
| without $E_{tx}$ | 86.50(±0.27) | 87.90(±0.31) |
| with $E_{tx}$ | **90.05(±0.19)** | **90.82(±0.18)** |

### A.5 Topics and Emotions Distribution of Cresci15

Figure 9 shows the distribution of topics and emotions with data labels in Cresci15,

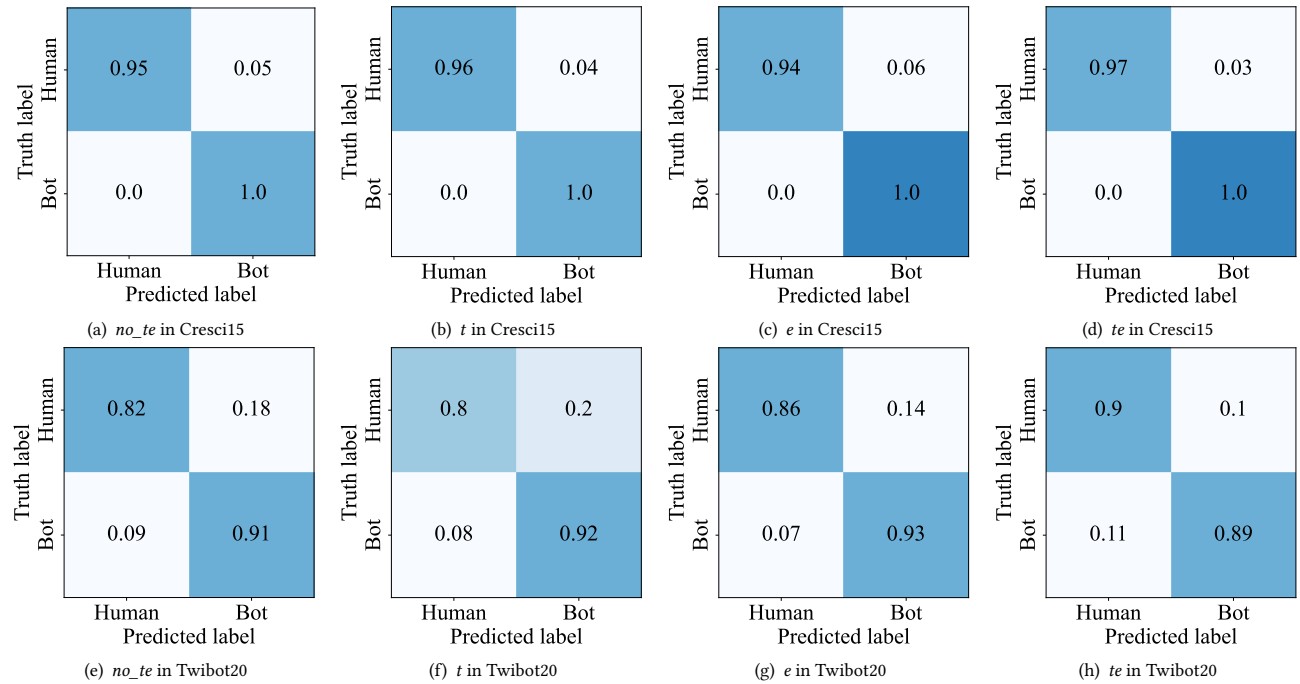

Figure 10: Confusion matrices for ETS-MM across four scenarios(*no_te*, *t*, *e* and *te*). The first four matrices correspond to the Cresci15 dataset, while the four matrices correspond to the Twibot20 dataset.

