# OpenReview forum: "ETS-MM: A Multi-Modal Social Bot Detection Model Based on Enhanced Textual Semantic Representation"
_ACM.org/TheWebConf/2025/Conference — WWW 2025 Poster_

### Official Review · Reviewer_oiMi · 2024-11-22

**Novelty:** 3
**Technical Quality:** 3

**Review:**

The authors introduce a new bot detection algorithm, that takes into account multiple inputs.
Overall, the concept and idea of the research make a lot of sense and are well-introduced. However, there are a few flaws throughout the manuscript which I see as fundamental.

Pros:
1. The usage of a multimodal approach for bot detection makes a lot of sense and can lead to better model performance.
2. Intro, motivation, and explanation of why targeting toward emotion and topics are well explained.

Cons:
1. There are many unclear definitions/terminology used in the manuscript. Some examples: what is the difference between LMs and LLMs? What is the exact definition of a social bot? Is it just a bot that works on a social network? Does it include sock puppets?
2. Looking at the results section, the results for the Cresci15 are really high. Is there any meaning to improving existing baselines of 99+ F1 scores? I would understand if the specific errors of existing models are critical for future usage of such systems, but it is not clear from the manuscript if this is the case.
3.  Cresci15 results - I understand that, on average, you are better than other models, but is this significant? Using the std you provide, this diff is not significant.
4. The whole evaluation is based on Twitter data. What about other platforms that have bots?
5. The motivation for the work is to incorporate specific aspects of the text into account (sentiment and topic). However, the whole algorithm is more complex than that, taking into account other aspects that make it hard to distinguish the added value of the original idea of the authors. Overall, the technical complexity is high, without too much explanation why.
6. Usage of zero-shot models is reasonable, however, for emotion and topics why not using state-of-the-are models that better perform on these tasks?

**Questions:**

1. Did you include distribution of the target feature for each dataset? I see some stats in Table 6, but I am curious what is the proportion of bots in each dataset.
2. Is there any reason to have the same colors in the figures for topics and emption types? If not, this is very confusing for the reader. I also recommend to add patterns to the bar plots for color blind readers.

**Reviewer Confidence:**

3: The reviewer is confident but not certain that the evaluation is correct

**Scope:**

4: The work is relevant to the Web and to the track, and is of broad interest to the community

---

### Official Review · Reviewer_6BEr · 2024-11-30

**Novelty:** 4
**Technical Quality:** 5

**Review:**

This study aims to develop a model for detecting social media bots by leveraging users' semantic feature information. The effectiveness of the proposed model is demonstrated through comparisons with existing models. The paper is well-structured and written clearly and understandably. The necessity and efficacy of the additional information incorporated into the model are appropriately discussed, and the model's validity is convincingly argued. The experiments and comparisons of the model also appear to be conducted appropriately.

Comments
1. The E_tx effect illustrated in Fig.7 does not seem to be very clear. Explaining how to interpret this figure and the basis for claiming that the separation between humans and bots is clear would be beneficial. Additional clarification on this point would help readers better understand the interpretation of Figure 2.

2. Including specific examples of bots that were detected by the proposed model but not by other models would make the advantages of the proposed model more apparent. Demonstrating these examples and their characteristics would further highlight the superiority of the proposed model.

**Questions:**

1. Why can it be clearly stated that there is an E_tx effect in Fig.7? Based on specific data or analysis results, please explain how the E_tx effect contributes to bot detection.

2. What are some specific examples or characteristics of bots that were detected by the proposed model but not by other models? Detailed case studies or characteristics would help clearly demonstrate the proposed model's effectiveness.

**Reviewer Confidence:**

2: The reviewer is willing to defend the evaluation, but it is likely that the reviewer did not understand parts of the paper

**Scope:**

4: The work is relevant to the Web and to the track, and is of broad interest to the community

---

### Official Review · Reviewer_bHcL · 2024-12-02

**Novelty:** 6
**Technical Quality:** 6

**Review:**

This manuscript introduces ETS-MM, a multimodal framework for detecting social bots in social networks. It augments multidimensional text-based information and extracts semantic features by analyzing tweeting behavior based on topic preference and emotion tendency. The framework integrates multimodal information to capture complex user interactions. Experimental results across two datasets demonstrate its superiority over existing methods.

Quality:
The proposed ETS-MM framework demonstrates a high level of technical sophistication. It combines multiple modalities of user data in a novel way to detect social bots, and the experimental results show significant improvement over existing methods. The methodology is well-explained, and the experimental setup is robust.

Clarity:
The paper is well-written and organized. The authors clearly explain the motivation behind the work, the methodology used, and the experimental results. The language is concise and accessible, making it easy to understand the contributions of the work. However, there are some errors in the text, such as incorrect use of single quotation marks and missing spaces between certain words.

Originality:
The work is original and introduces a novel approach to detecting social bots. The integration of topic preference, emotion tendency, and multimodal information is a unique aspect of the proposed framework that sets it apart from existing methods.

Significance:
The significance of this work lies in its potential to improve the security and authenticity of social media platforms by detecting social bots more effectively. The proposed method could have practical implications for social media companies and researchers in the field of artificial intelligence and security.

Pros:

The proposed ETS-MM framework is technically sophisticated and combines multiple modalities of user data in a novel way.
The experimental results show significant improvement over existing methods.
The paper is well-written and organized, making it easy to understand the contributions of the work.
The work is original and introduces a unique approach to detecting social bots.

Cons:

The authors do not discuss potential limitations related to the proposed framework.
There are some errors in the text, such as incorrect use of single quotation marks and missing spaces between certain words.

Overall, this work represents a significant contribution to the field of social bot detection and demonstrates the potential of multimodal approaches in improving the accuracy and effectiveness of detection methods.

**Questions:**

The explanation provided for Figure 8 in the following text is confusing. Please elaborate further.

In addition, we extracted five types of features for generating user
embeddings across two GNN convolution layers and visualized
them using t-SNE, as shown in Figure 8. The t-SNE visualization
clearly shows that different features contribute to distinct clustering patterns. In both datasets, these modalities integrate with one
another. Particularly in the Cresci15 dataset, feature points from
different modalities gradually merge into the same category cluster,
further validating the effectiveness of our proposed method

**Reviewer Confidence:**

4: The reviewer is certain that the evaluation is correct and very familiar with the relevant literature

**Scope:**

4: The work is relevant to the Web and to the track, and is of broad interest to the community

---

### Official Review · Reviewer_cZ5y · 2024-12-02

**Novelty:** 4
**Technical Quality:** 4

**Review:**

This work introduces ETS-MM, a multimodal social bot detection framework that leverages topic preferences and emotional tendencies to enhance semantic feature representation. By analyzing users’ tweeting behavior, the framework integrates topics and emotions into tweet data, creating hybrid textual sequences. A follow-up language model (LM) captures the relationships between content and behavior, while a graph neural network (GNN) combines metadata, textual features, semantic insights, and social network relationships for robust detection. While the method demonstrates promising performance and achieves state-of-the-art results on benchmark datasets, several areas for improvement remain:

(1) The novelty primarily lies in the combination of existing techniques rather than the introduction of fundamentally new methodologies. While the integration of topic preference and emotional tendency analysis to enhance semantic feature representation is creative, these components are not entirely new in the field. The framework still relies heavily on established methods.

(2) Although the use of LLMs and GNNs is effective, it presents significant computational challenges, particularly for large-scale datasets. This limitation may hinder the framework’s scalability and its practical application to extensive social media platforms.

(3) The framework’s reliance on analyzing user metadata, textual content, and social network interactions raises important privacy and ethical concerns in real-world applications, using online LLMs for data analysis. Open-source LLMs may offer a viable alternative to address these issues.

In summary, the paper provides a valuable contribution by achieving high performance in social bot detection through multimodal integration. However, its reliance on existing techniques, scalability limitations, and unresolved ethical considerations highlight areas that require further refinement to ensure broader applicability and responsible use. Additionally, a minor comment: there are several places where spaces are missing.

**Questions:**

1. As shown in Table 3, the combination of various LMs and GNNs can lead to differing performance outcomes. Which combination was reported in Table 2?
2. For example, the emotion analysis for the topic "news" appears to differ between Cresci15 and Twibot20. Could this topic-emotion difference be too specific to the datasets, lacking a consistent pattern across different datasets? If possible, adding additional data to evaluate the model's performance could provide more comprehensive insights.

**Reviewer Confidence:**

4: The reviewer is certain that the evaluation is correct and very familiar with the relevant literature

**Scope:**

4: The work is relevant to the Web and to the track, and is of broad interest to the community